# Stacked Capsule Autoencoders

**Adam R. Kosiorek** [*†‡]
adamk@robots.ox.ac.uk

**Sara Sabour** [§]  **Yee Whye Teh** [∇]  **Geoffrey E. Hinton** [§]

[‡] **Applied AI Lab**
Oxford Robotics Institute
University of Oxford

[†] **Department of Statistics**
University of Oxford

[§] **Google Brain**
Toronto

[∇] **DeepMind**
London

## Abstract

Objects are composed of a set of geometrically organized parts. We introduce an unsupervised capsule autoencoder (SCAE), which explicitly uses geometric relationships between parts to reason about objects. Since these relationships do not depend on the viewpoint, our model is robust to viewpoint changes. SCAE consists of two stages. In the first stage, the model predicts presences and poses of part templates directly from the image and tries to reconstruct the image by appropriately arranging the templates. In the second stage, SCAE predicts parameters of a few object capsules, which are then used to reconstruct part poses. Inference in this model is amortized and performed by off-the-shelf neural encoders, unlike in previous capsule networks. We find that object capsule presences are highly informative of the object class, which leads to state-of-the-art results for unsupervised classification on SVHN (55%) and MNIST (98.7%).

## 1   Introduction

Convolutional neural networks (CNN) work better than networks without weight-sharing because of their inductive bias: if a local feature is useful in one image location, the same feature is likely to be useful in other locations. It is tempting to exploit other effects of viewpoint changes by replicating features across scale, orientation and other affine degrees of freedom, but this quickly leads to cumbersome, high-dimensional feature maps.

An alternative to replicating features across the non-translational degrees of freedom is to explicitly learn transformations between the natural coordinate frame of a whole object and the natural coordinate frames of each of its parts. Computer graphics relies on such object→part coordinate transformations to represent the geometry of an object in a viewpoint-invariant manner. Moreover, there is strong evidence that, unlike standard CNNs, human vision also relies on coordinate frames: imposing an unfamiliar coordinate frame on a familiar object makes it challenging to recognize the object or its geometry (Rock, 1973; G. E. Hinton, 1979).

A neural system can learn to reason about transformations between objects, their parts and the viewer, but each kind of transformation will likely need to be represented differently. An object-part-relationship (OP) is viewpoint-invariant, approximately constant and could be easily coded by learned weights. The relative coordinates

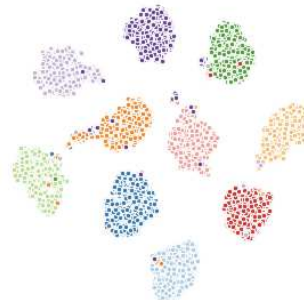

Figure 1: SCAEs learn to explain different object classes with separate object capsules, thereby doing unsupervised classification. Here, we show TSNE embeddings of object capsule presence probabilities for 10000 MNIST digits. Individual points are color-coded according to the corresponding digit class.

---

[*]This work was done during an internship at Google Brain.

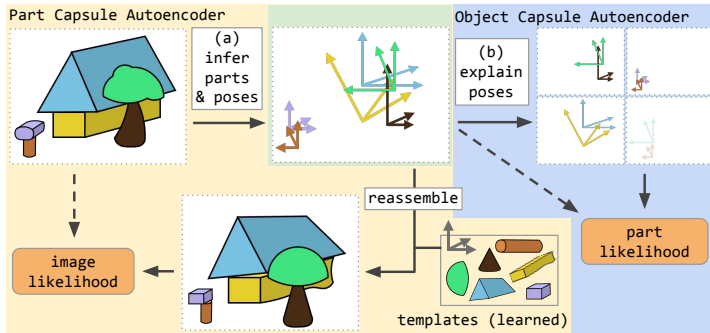

Figure 2: Stacked Capsule Autoencoder (SCAE): (a) *part* capsules segment the input into parts and their poses. The poses are then used to reconstruct the input by affine-transforming learned templates. (b) *object* capsules try to arrange inferred poses into objects, thereby discovering underlying structure. SCAE is trained by maximizing image and part log-likelihoods subject to sparsity constraints.

of an object (or a part) with respect to the viewer change with the viewpoint (they are viewpoint-equivariant), and could be easily coded with neural activations[2].

With this representation, the pose of a single object is represented by its relationship to the viewer. Consequently, representing a single object does not necessitate replicating neural activations across space, unlike in CNNs. It is only processing two (or more) different instances of the same type of object in parallel that requires spatial replicas of both model parameters and neural activations.

In this paper we propose the Stacked Capsule Autoencoder (SCAE), which has two stages (Fig. 2). The first stage, the Part Capsule Autoencoder (PCAE), segments an image into constituent parts, infers their poses, and reconstructs the image by appropriately arranging affine-transformed part templates. The second stage, the Object Capsule Autoencoder (OCAE), tries to organize discovered parts and their poses into a smaller set of objects. These objects then try to reconstruct the part poses using a separate mixture of predictions for each part. Every object capsule contributes components to each of these mixtures by multiplying its pose—the object-viewer-relationship (OV)—by the relevant object-part-relationship (OP).

Stacked Capsule Autoencoders (Section 2) capture spatial relationships between whole objects and their parts when trained on unlabelled data. The vectors of presence probabilities for the object capsules tend to form tight clusters (cf. Figure 1), and when we assign a class to each cluster we achieve state-of-the-art results for unsupervised classification on SVHN (55%) and MNIST (98.7%), which can be further improved to 67% and 99%, respectively, by learning fewer than 300 parameters (Section 3). We describe related work in Section 4 and discuss implications of our work and future directions in Section 5. The code is available at github.com/google-research/google-research/tree/master/stacked_capsule_autoencoders.

## 2 Stacked Capsule Autoencoders (SCAE)

Segmenting an image into parts is non-trivial, so we begin by abstracting away pixels and the part-discovery stage, and develop the Constellation Capsule Autoencoder (CCAE) (Section 2.1). It uses two-dimensional points as parts, and their coordinates are given as the input to the system. CCAE learns to model sets of points as arrangements of familiar constellations, each of which has been transformed by an independent similarity transform. The CCAE learns to assign individual points to their respective constellations—without knowing the number of constellations or their shapes in advance. Next, in Section 2.2, we develop the Part Capsule Autoencoder (PCAE) which learns to infer parts and their poses from images. Finally, we stack the Object Capsule Autoencoder (OCAE), which closely resembles the CCAE, on top of the PCAE to form the Stacked Capsule Autoencoder (SCAE).

### 2.1 Constellation Autoencoder (CCAE)

Let $\{\mathbf{x}_m \mid m = 1, \ldots, M\}$ be a set of two-dimensional input points, where every point belongs to a constellation as in Figure 3. We first encode all input points (which take the role of part capsules) with Set Transformer (Lee et al., 2019)—a permutation-invariant encoder $h^{\text{caps}}$ based on attention mechanisms—into $K$ object capsules. An object capsule $k$ consists of a capsule feature vector $\mathbf{c}_k$, its presence probability $a_k \in [0, 1]$ and a $3 \times 3$ object-viewer-relationship (OV) matrix, which represents

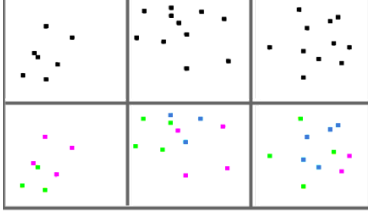

Figure 3: Unsupervised segmentation of points belonging to up to three constellations of squares and triangles at different positions, scales and orientations. The model is trained to reconstruct the points (top row) under the CCAE mixture model. The bottom row colours the points based on the parent with the highest posterior probability in the mixture model. The right-most column shows a failure case. Note that the model uses sets of points, not pixels, as its input; we use images only to visualize the constellation arrangements.

the affine transformation between the object (constellation) and the viewer. Note that each object capsule can represent only one object at a time. Every object capsule uses a separate multilayer perceptron (MLP) $\mathrm{h}_k^{\mathrm{part}}$ to predict $N \leq M$ part candidates from the capsule feature vector $\mathbf{c}_k$. Each candidate consists of the conditional probability $a_{k,n} \in [0,1]$ that a given candidate part exists, an associated scalar standard deviation $\lambda_{k,n}$, and a $3 \times 3$ object-part-relationship (OP) matrix, which represents the affine transformation between the object capsule and the candidate part[3]. Candidate predictions $\mu_{k,n}$ are given by the product of the object capsule OV and the candidate OP matrices. We model all input points as a single Gaussian mixture, where $\mu_{k,n}$ and $\lambda_{k,n}$ are the centres and standard deviations of the isotropic Gaussian components. See Figures 2 and 6 for illustration; formal description follows:

$$\mathrm{OV}_{1:K}, \mathbf{c}_{1:K}, a_{1:K} = \mathrm{h}^{\mathrm{caps}}(\mathbf{x}_{1:M}) \qquad \text{predict object capsule parameters,} \qquad (1)$$

$$\mathrm{OP}_{k,1:N}, a_{k,1:N}, \lambda_{k,1:N} = \mathrm{h}_k^{\mathrm{part}}(\mathbf{c}_k) \qquad \text{decode candidate parameters from } c_k\text{'s,} \qquad (2)$$

$$V_{k,n} = \mathrm{OV}_k \mathrm{OP}_{k,n} \qquad \text{decode a part pose candidate,} \qquad (3)$$

$$p(\mathbf{x}_m \mid k, n) = \mathcal{N}(\mathbf{x}_m \mid \mu_{k,n}, \lambda_{k,n}) \qquad \text{turn candidates into mixture components,} \qquad (4)$$

$$p(\mathbf{x}_{1:M}) = \prod_{m=1}^{M} \sum_{k=1}^{K} \sum_{n=1}^{N} \frac{a_k a_{k,n}}{\sum_i a_i \sum_j a_{i,j}} \, p(\mathbf{x}_m \mid k, n) \,. \qquad (5)$$

The model is trained without supervision by maximizing the likelihood of part capsules in Equation (5) subject to sparsity constraints, *cf.* Section 2.4 and Appendix C. The part capsule $m$ can be assigned to the object capsule $k^\star$ by looking at the mixture component responsibility, that is $k^\star = \arg\max_k \ a_k a_{k,n} \ p(\mathbf{x}_m \mid k, n)$.[4] Empirical results show that this model is able to perform unsupervised instance-level segmentation of points belonging to different constellations, even in data which is difficult to interpret for humans. See Figure 3 for an example and Section 3.1 for details.

## 2.2 Part Capsule Autoencoder (PCAE)

Explaining images as geometrical arrangements of parts requires 1) discovering what parts are there in an image and 2) inferring the relationships of the parts to the viewer (their pose). For the CCAE a part is just a 2D point (that is, a (x, y) coordinate), but for the PCAE each part capsule $m$ has a six-dimensional pose $\mathbf{x}_m$ (two rotations, two translations, scale and shear), a presence variable $d_m \in [0,1]$ and a unique identity. We frame the part-discovery problem as auto-encoding: the encoder learns to infer the poses and presences of different part capsules, while the decoder learns an image template $T_m$ for each part (Fig. 4) similar to Tieleman, 2014; Eslami et al., 2016. If a part exists (according to its presence variable), the corresponding template is affine-transformed with the inferred pose giving $\widehat{T}_m$. Finally, transformed templates are arranged into the image. The PCAE is followed by an Object Capsule Autoencoder (OCAE), which closely resembles the CCAE and is described in Section 2.3.

Let $\mathbf{y} \in [0,1]^{h \times w \times c}$ be the image. We limit the maximum number of part capsules to $M$ and use an encoder to infer their poses $\mathbf{x}_m$, presence probabilities $d_m$, and special features $\mathbf{z}_m \in \mathbb{R}^{c_z}$, one per part capsule. Special features can be used to alter the templates in an input-dependent manner (we use them to predict colour, but more complicated mappings are possible). The special features also inform the OCAE about unique aspects of the corresponding part (e. g., occlusion or relation to other parts). Templates $T_m \in [0,1]^{h_t \times w_t \times (c+1)}$ are smaller than the image $\mathbf{y}$, but have an additional alpha

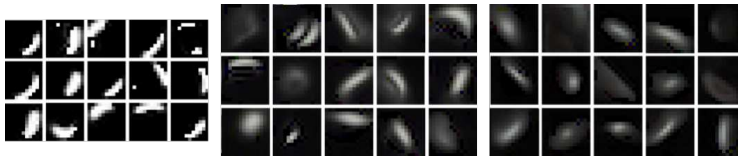

Figure 4: Stroke-like templates learned on MNIST (left) as well as sobel-filtered SVHN (middle) and CIFAR10 (right). For SVHN they often take the form of double strokes due to sobel filtering.

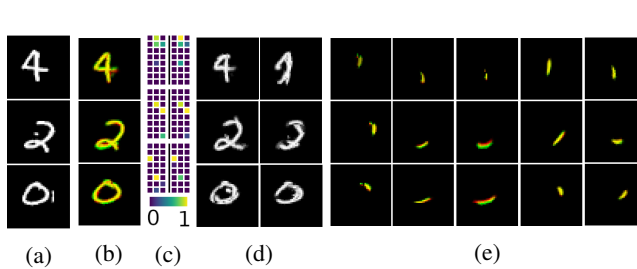

(a) (b) (c) (d) (e)

Figure 5: MNIST (a) images, (b) reconstructions from part capsules in red and object capsules in green, with overlapping regions in yellow. Only a few object capsules are activated for every input (c) a priori (left) and even fewer are needed to reconstruct it (right). The most active capsules (d) capture object identity and its appearance; (e) shows a few o f the affine-transformed templates used for reconstruction.

channel which allows occlusion by other templates. We use $T_m^a$ to refer to the alpha channel and $T_m^c$ to refer to its colours.

We allow each part capsule to be used only once to reconstruct an image, which means that parts of the same type are not repeated[5]. To infer part capsule parameters we use a CNN-based encoder followed by *attention-based pooling*, which is described in more detail in the Appendix E and whose effects on the model performance are analyzed in Section 3.3.

The image is modelled as a spatial Gaussian mixture, similarly to Greff et al., 2019; Burgess et al., 2019; Engelcke et al., 2019. Our approach differs in that we use pixels of the transformed templates (instead of component-wise reconstructions) as the centres of isotropic Gaussian components, but we also use constant variance. Mixing probabilities of different components are proportional to the product of presence probabilities of part capsules and the value of the learned alpha channel for every template. More formally:

$$\mathbf{x}_{1:M}, d_{1:M}, \mathbf{z}_{1:M} = \mathrm{h}^{\mathrm{enc}}(\mathbf{y}) \qquad \text{predict part capsule parameters,} \qquad (6)$$

$$\boldsymbol{c}_m = \mathrm{MLP}(\mathbf{z}_m) \qquad \text{predict the color of the m}^{\mathrm{th}} \text{ template,} \qquad (7)$$

$$\widehat{T}_m = \mathrm{TransformImage}(T_m, \mathbf{x}_m) \qquad \text{apply affine transforms to image templates,} \qquad (8)$$

$$p_{m,i,j}^y \propto d_m \widehat{T}_{m,i,j}^a \qquad \text{compute mixing probabilities,} \qquad (9)$$

$$p(\mathbf{y}) = \prod_{i,j} \sum_{m=1}^{M} p_{m,i,j}^y \mathcal{N}\left(y_{i,j} \mid \boldsymbol{c}_m \cdot \widehat{T}_{m,i,j}^c; \sigma_y^2\right) \qquad \text{calculate the image likelihood.} \qquad (10)$$

Training the PCAE results in learning templates for object parts, which resemble strokes in the case of MNIST, see Figure 4. This stage of the model is trained by maximizing the image likelihood of Equation (10).

## 2.3 Object Capsule Autoencoder (OCAE)

Having identified parts and their parameters, we would like to discover objects that could be composed of them[6]. To do so, we use concatenated poses $\mathbf{x}_m$, special features $\mathbf{z}_m$ and flattened templates $T_m$ (which convey the identity of the part capsule) as an input to the OCAE, which differs from the CCAE in the following ways. Firstly, we feed part capsule presence probabilities $d_m$ into the OCAE's encoder—these are used to bias the Set Transformer's attention mechanism not to take absent points into account. Secondly, $d_m$s are also used to weigh the part-capsules' log-likelihood, so that we do not take log-likelihood of absent points into account. This is implemented by raising the likelihood of the $m^{\mathrm{th}}$ part capsule to the power of $d_m$, *cf.* Equation (5). Additionally, we stop the gradient on all of OCAE's inputs except the special features to improve training stability and avoid the problem

of collapsing latent variables; see e. g., Rasmus et al., 2015. Finally, parts discovered by the PCAE have independent identities (templates and special features rather than 2D points). Therefore, every part-pose is explained as an independent mixture of predictions from object-capsules—where every object capsule makes exactly $M$ candidate predictions $V_{k,1:M}$, or exactly **one** candidate prediction per part. Consequently, the part-capsule likelihood is given by,

$$p(\mathbf{x}_{1:M}, d_{1:M}) = \prod_{m=1}^{M} \left[ \sum_{k=1}^{K} \frac{a_k a_{k,m}}{\sum_i a_i \sum_j a_{i,j}} \, p(\mathbf{x}_m \mid k, m) \right]^{d_m} . \tag{11}$$

The OCAE is trained by maximising the part pose likelihood of Equation (11), and it learns to discover further structure in previously identified parts, leading to learning sparsely-activated object capsules, see Figure 5. Achieving this sparsity requires further regularization, however.

## 2.4 Achieving Sparse and Diverse Capsule Presences

Stacked Capsule Autoencoders are trained to maximise pixel and part log-likelihoods ($\mathcal{L}_{ll} = \log p(\mathbf{y}) + \log p(\mathbf{x}_{1:M})$). If not constrained, however, they tend to either use all of the part and object capsules to explain every data example or collapse onto always using the same subset of capsules, regardless of the input. We want the model to use different sets of part-capsules for different input examples and to specialize object-capsules to particular arrangements of parts. To encourage this, we impose sparsity and entropy constraints. We evaluate their importance in Section 3.3.

We first define prior and posterior object-capsule presence as follows. For a minibatch of size $B$ with $K$ object capsules and $M$ part capsules we define a minibatch of prior capsule presence $a_{1:K}^{\mathrm{prior}}$ with dimension $[B, K]$ and posterior capsule presence $a_{1:K,1:M}^{\mathrm{posterior}}$ with dimension $[B, K, M]$ as,

$$a_k^{\mathrm{prior}} = a_k \max_m a_{m,k} , \qquad a_{k,m}^{\mathrm{posterior}} = a_k a_{k,m} \, \mathcal{N}(\mathbf{x}_m \mid m, k) , \tag{12}$$

respectively; the former is the maximum presence probability among predictions from object capsule $k$ while the latter is the unnormalized mixing proportion used to explain part capsule $m$.

**Prior sparsity** Let $\overline{u}_k = \sum_{b=1}^{B} a_{b,k}^{\mathrm{prior}}$ the sum of presence probabilities of the object capsule $k$ among different training examples, and $\widehat{u}_b = \sum_{k=1}^{K} a_{b,k}^{\mathrm{prior}}$ the sum of object capsule presence probabilities for a given example. If we assume that training examples contain objects from different classes uniformly at random and we would like to assign the same number of object capsules to every class, then each class would obtain $K/C$ capsules. Moreover, if we assume that only one object is present in every image, then $K/C$ object capsules should be present for every input example, which results in the sum of presence probabilities of $B/C$ for every object capsule. To this end, we minimize,

$$\mathcal{L}_{\mathrm{prior}} = \frac{1}{B} \sum_{b=1}^{B} \left( \widehat{u}_b - K/C \right)^2 + \frac{1}{K} \sum_{k=1}^{K} \left( \overline{u}_k - B/C \right)^2 . \tag{13}$$

**Posterior Sparsity** Similarly, we experimented with minimizing the within-example entropy of capsule posterior presence $\mathcal{H}(\overline{v}_k)$ and maximizing its between-example entropy $\mathcal{H}(\widehat{v}_b)$, where $\mathcal{H}$ is the entropy, and where $\overline{v}_k$ and $\widehat{v}_b$ are the the normalized versions of $\sum_{k,m} a_{b,k,m}^{\mathrm{posterior}}$ and $\sum_{b,m} a_{b,k,m}^{\mathrm{posterior}}$, respectively. The final loss reads as

$$\mathcal{L}_{\mathrm{posterior}} = \frac{1}{K} \sum_{k=1}^{K} \mathcal{H}(\overline{v}_k) - \frac{1}{B} \sum_{b=1}^{B} \mathcal{H}(\widehat{v}_b) . \tag{14}$$

Our ablation study has shown, however, that the model can perform equally well without these posterior sparsity constraints, cf. Section 3.3.

Fig. 6 shows the schematic architecture of SCAE. We optimize a weighted sum of image and part likelihoods and the auxiliary losses. Loss weight selection process, as well as the values used for experiments, are detailed in Appendix A.

In order to make the values of presence probabilities ($a_k, a_{k,m}$ and $d_m$) closer to binary we inject uniform noise $\in [-2, 2]$ into logits, similar to Tieleman, 2014. This forces the model to predict logits that are far from zero to avoid stochasticity and makes the predicted presence probabilities close to binary. Interestingly, it tends to work better in our case than using the Concrete distribution (Maddison et al., 2017).

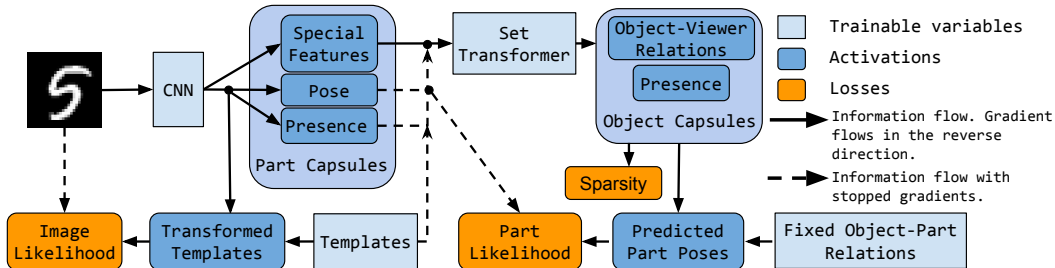

Figure 6: SCAE architecture.

# 3 Evaluation

The decoders in the SCAE use explicitly parameterised affine transformations that allow the encoders' inputs to be explained with a small set of transformed objects or parts. The following evaluations show how the embedded geometrical knowledge helps to discover patterns in data. Firstly, we show that the CCAE discovers underlying structures in sets of two-dimensional points, thereby performing instance-level segmentation. Secondly, we pair an OCAE with a PCAE and investigate whether the resulting SCAE can discover structure in real images. Finally, we present an ablation study that shows which components of the model contribute to the results.

## 3.1 Discovering Constellations

We create arrangements of constellations online, where every input example consists of up to 11 two-dimensional points belonging to up to three different constellations (two squares and a triangle) as well as binary variables indicating the presence of the points (points can be missing). Each constellation is included with probability $0.5$ and undergoes a similarity transformation, whereby it is randomly scaled, rotated by up to $180°$ and shifted. Finally, every input example is normalised such that all points lie within $[-1, 1]^2$. Note that we use sets of points, and not images, as inputs to our model.

We compare the CCAE against a baseline that uses the same encoder but a simpler decoder: the decoder uses the capsule parameter vector $\mathbf{c}_k$ to directly predict the location, precision and presence probability of each of the four points as well as the presence probability of the whole corresponding constellation. Implementation details are listed in Appendix A.1.

Both models are trained unsupervised by maximising the part log-likelihood. We evaluate them by trying to assign each input point to one of the object capsules. To do so, we assign every input point to the object capsule with the highest posterior probability for this point, *cf.* Section 2.1, and compute segmentation accuracy (i. e., the true-positive rate).

The CCAE consistently achieves[7] below $4\%$ error with the best model achieving $2.8\%$ , while the best baseline achieved $26\%$ error using the same budget for hyperparameter search. This shows that wiring in an inductive bias towards modelling geometric relationships can help to bring down the error by an order of magnitude—at least in a toy setup where each set of points is composed of familiar constellations that have been independently transformed.

## 3.2 Unsupervised Class Discovery in Images

We now turn to images in order to assess if our model can simultaneously learn to discover parts and group them into objects. To allow for multimodality in the appearance of objects of a specific class, we typically use more object capsules than the number of class labels. It turns out that the vectors of presence probabilities form tight clusters as shown by their TSNE embeddings (Maaten and G. E. Hinton, 2008) in Figure 1—note the large separation between clusters corresponding to different digits, and that only a few data points are assigned to the wrong clusters. Therefore, we expect object capsules presences to be highly informative of the class label. To test this hypothesis,

Table 1: Unsupervised classification results in % with (standard deviation) are averaged over 5 runs. Methods based on mutual information are shaded. Results marked with † use data augmentation, ∇ use IM-AGENET-pretrained features instead of images, while § are taken from Ji et al., 2018. We highlight the best results and those that are are within its 98% confidence interval according to a two-sided t test.

| Method | MNIST | CIFAR10 | SVHN |
|---|---|---|---|
| KMEANS (Haeusser et al., 2018) | 53.49 | 20.8 | 12.5 |
| AE (Bengio et al., 2007)[§] | 81.2 | 31.4 | - |
| GAN (Radford et al., 2016)[§] | 82.8 | 31.5 | - |
| IMSAT (Hu et al., 2017)[†,∇] | **98.4** (0.4) | 45.6 (0.8) | **57.3** (3.9) |
| IIC (Ji et al., 2018)[§,†] | **98.4** (0.6) | **57.6** (5.0) | - |
| ADC (Haeusser et al., 2018)[†] | **98.7** (0.6) | 29.3 (1.5) | 38.6 (4.1) |
| SCAE (LIN-MATCH) | **98.7 (0.35)** | 25.01 (1.0) | **55.33 (3.4)** |
| SCAE (LIN-PRED) | **99.0 (0.07)** | 33.48 (0.3) | **67.27 (4.5)** |

we train SCAE on MNIST, SVHN[8] and CIFAR10 and try to assign class labels to vectors of object capsule presences. This is done with one of the following methods: LIN-MATCH: after finding 10 clusters[9] with KMEANS we use bipartite graph matching (Kuhn, 1955) to find the permutation of cluster indices that minimizes the classification error—this is standard practice in unsupervised classification, see e. g., Ji et al., 2018; LIN-PRED: we train a linear classifier with supervision given the presence vectors; this learns $K \times 10$ weights and 10 biases, where $K$ is the number of object capsules, but it does not modify any parameters of the main model.

In agreement with previous work on unsupervised clustering (Ji et al., 2018; Hu et al., 2017; Hjelm et al., 2019; Haeusser et al., 2018), we train our models and report results on full datasets (TRAIN, VALID and TEST splits). The linear transformation used in LIN-PRED variant of our method is trained on the TRAIN split of respective datasets while its performance on the TEST split is reported.

We used an PCAE with 24 single-channel $11 \times 11$ templates for MNIST and 24 and 32 three-channel $14 \times 14$ templates for SVHN and CIFAR10, respectively. We used sobel-filtered images as the reconstruction target for SVHN and CIFAR10, as in Jaiswal et al., 2018, while using the raw pixel intensities as the input to PCAE. The OCAE used 24, 32 and 64 object capsules, respectively. Further details on model architectures and hyper-parameter tuning are available in Appendix A. All results are presented in Table 1. SCAE achieves state-of-the-art results in unsupervised object classification on MNIST and SVHN and under-performs on CIFAR10 due to the inability to model backgrounds, which is further discussed in Section 5.

### 3.3 Ablation study

SCAEs have many moving parts; an ablation study shows which model components are important and to what degree. We train SCAE variants on MNIST as well as a padded-and-translated $40 \times 40$ version of the dataset, where the original digits are translated up to 6 pixels in each direction. Trained models are tested on TEST splits of both datasets; additionally, we evaluate the model trained on the $40 \times 40$ MNIST on the TEST split of AFFNIST dataset. Testing on AFFNIST shows whether the model can generalise to unseen viewpoints. This task was used by Rawlinson et al., 2018 to evaluate Sparse Unsupervised Capsules, which achieved $90.12\%$ accuracy. SCAE achieves $92.2 \pm 0.59\%$, which indicates that it is better at viewpoint generalisation. We choose the LIN-MATCH performance metric, since it is the one favoured by the unsupervised classification community.

Results are split into several groups and shown in Table 2. We describe each group in turn. Group a) shows that sparsity losses introduced in Section 2.4 increase model performance, but that the posterior loss might not be necessary. Group b) checks the influence of injecting noise into logits for presence probabilities, *cf.* Section 2.4. Injecting noise into part capsules seems critical, while noise in object capsules seems unnecessary—the latter might be due to sparsity losses. Group c) shows that using similarity (as opposed to affine) transforms in the decoder can be restrictive in some cases, while not allowing deformations hurts performance in every case.

Group d) evaluates the type of the part-capsule encoder. The LINEAR encoder entails a CNN followed by a fully-connected layer, while the CONV encoder predicts one feature map for every capsule parameter, followed by global-average pooling. The choice of part-capsule encoder seems

Table 2: Ablation study on MNIST. All used model components contribute to its final performance. AFFNIST results show out-of-distribution generalization properties and come from a model trained on $40 \times 40$ MNIST. Numbers represent average % and (standard deviation) over 10 runs. We highlight the best results and those that are are within its 98% confidence interval according to a two-sided t test.

| | Method | MNIST | $40 \times 40$ MNIST | AFFNIST |
|---|---|---|---|---|
| | full model | **95.3 (4.65)** | **98.7 (0.35)** | **92.2 (0.59)** |
| a) | no posterior sparsity | **97.5 (1.55)** | **95.0 (7.20)** | **85.3 (11.67)** |
| | no prior sparsity | 72.4 (22.39) | 88.2 (6.98) | 71.3 (5.46) |
| | no prior/posterior sparsity | 84.7 (3.01) | 82.0 (5.46) | 59.0 (5.66) |
| b) | no noise in object caps | **96.7 (2.30)** | **98.5 (0.12)** | **93.5 (0.38)** |
| | no noise in any caps | **93.1 (5.09)** | 78.5 (22.69) | 64.1 (26.74) |
| | no noise in part caps | **93.9 (7.16)** | 82.8 (24.83) | 70.7 (25.96) |
| c) | similarity transforms | **97.5 (1.55)** | 95.9 (1.59) | 88.9 (1.58) |
| | no deformations | 87.3 (21.48) | 87.2 (18.54) | 79.0 (22.44) |
| d) | LINEAR part enc | **98.0 (0.52)** | 63.2 (31.47) | 50.8 (26.46) |
| | CONV part enc | **97.6 (1.22)** | **97.8 (.98)** | 81.6 (1.66) |
| e) | MLP enc for object caps | 27.1 (9.03) | 36.3 (3.70) | 25.29 (3.69) |
| f) | no special features | 90.7 (2.25) | 58.7 (31.60) | 44.5 (21.71) |

not to matter much for within-distribution performance; however, our attention-based pooling (cf. Appendix E) does achieve much higher classification accuracy when evaluated on a different dataset, showing better generalisation to novel viewpoints.

Additionally, e) using Set Transformer as the object-capsule encoder is essential. We hypothesise that it is due to the natural tendency of Set Transformer to find clusters, as reported in Lee et al., 2019. Finally, f) using special features $\mathbf{z}_m$ seems not less important—presumably due to effects the high-level capsules have on the representation learned by the primary encoder.

## 4 Related Work

**Capsule Networks** Our work combines ideas from Transforming Autoencoders (G. E. Hinton, Krizhevsky, et al., 2011) and EM Capsules (G. E. Hinton, Sabour, et al., 2018). Transforming autoencoders discover affine-aware capsule *instantiation parameters* by training an autoencoder to reconstruct an affine-transformed version of the original image. This model uses an additional input that explicitly represents the transformation, which is a form of supervision. By contrast, our model does not need any input other than the image.

Both EM Capsules and the preceding Dynamic Capsules (Sabour et al., 2017) use the poses of parts and learned part→object relationships to vote for the poses of objects. When multiple parts cast very similar votes, the object is assumed to be present, which is facilitated by an interactive inference (routing) algorithm. Iterative routing is inefficient and has prompted further research (Wang and Liu, 2018; Zhang et al., 2018; Li et al., 2018). In contrast to prior work, we use objects to predict parts rather than vice-versa; therefore we can dispense with iterative routing at inference time—every part is explained as a mixture of predictions from different objects, and can have only one parent. This regularizes the OCAE's encoder to respect the single parent constraint when learning to group parts into objects.

Additionally, since it is the objects that predict parts, the part poses can have fewer degrees-of-freedom than object poses (as in the CCAE). Inference is still possible because the OCAE encoder makes object predictions based on *all* the parts. This is in contrast to each individual part making its own prediction, as was the case in previous works on capsules.

A further advantage of our version of capsules is that it can perform unsupervised learning, whereas previous capsule networks used discriminative learning. Rawlinson et al., 2018 is a notable exception and used the reconstruction MLP introduced in Sabour et al., 2017 to train Dynamic Capsules without supervision. Their results show that unsupervised training for capsule-conditioned reconstruction helps with generalization to AFFNIST classification; we further improve on their results, *cf.* Section 3.3.

**Unsupervised Classification** There are two main approaches to unsupervised object category detection in computer vision. The first one is based on representation learning and typically requires discovering clusters or learning a classifier on top of the learned representation. Eslami et al., 2016; Kosiorek et al., 2018 use an iterative procedure to infer a variable number of latent variables, one for every object in a scene, that are highly informative of the object class, while Greff et al., 2019; Burgess et al., 2019 perform unsupervised instance-level segmentation in an iterative fashion. While

similar to our work, these approaches cannot decompose objects into their constituent parts and do not provide an explicit description of object shape (e. g., templates and their poses in our model).

The second approach targets classification explicitly by minimizing mutual information (MI)-based losses and directly learning class-assignment probabilities. IIC (Ji et al., 2018) maximizes an exact estimator of MI between two discrete probability vectors describing (transformed) versions of the input image. DeepInfoMax (Hjelm et al., 2019) relies on negative samples and maximizes MI between the predicted probability vector and its input via noise-contrastive estimation (Gutmann and Hyvärinen, 2010). This class of methods directly maximizes the amount of information contained in an assignment to discrete clusters, and they hold state-of-the-art results on most unsupervised classification tasks. MI-based methods suffer from typical drawbacks of mutual information estimation: they require massive data augmentation and large batch sizes. This is in contrast to our method, which achieves comparable performance with batch size no bigger than 128 and with no data augmentation.

**Geometrical Reasoning**   Other attempts at incorporating geometrical knowledge into neural networks include exploiting equivariance properties of group transformations (Cohen and Welling, 2016) or new types of convolutional filters (Oyallon and Mallat, 2015; Dieleman et al., 2016). Although they achieve significant parameter efficiency in handling rotations or reflections compared to standard CNNs, these methods cannot handle additional degrees of freedom of affine transformations—like scale. Lenssen et al., 2018 combined capsule networks with group convolutions to guarantee equivariance and invariance in capsule networks. Spatial Transformers (ST; Jaderberg et al., 2015) apply affine transformations to the image sampling grid while steerable networks (Cohen and Welling, 2017; Jacobsen et al., 2017) dynamically change convolutional filters. These methods are similar to ours in the sense that transformation parameters are predicted by a neural network but differ in the sense that ST uses global transformations applied to the whole image while steerable networks use only local transformations. Our approach can use different global transformations for every object as well as local transformations for each of their parts.

## 5   Discussion

The main contribution of our work is a novel method for representation learning, in which highly structured decoder networks are used to train one encoder network that can segment an image into parts and their poses and another encoder network that can compose the parts into coherent wholes. Even though our training objective is not concerned with classification or clustering, SCAE is the only method that achieves competitive results in unsupervised object classification without relying on mutual information (MI). This is significant since, unlike our method, MI-based methods require sophisticated data augmentation. It may be possible to further improve results by using an MI-based loss to train SCAE, where the vector of capsule probabilities could take the role of discrete probability vectors in IIC (Ji et al., 2018). SCAE under-performs on CIFAR10, which could be because of using fixed templates, which are not expressive enough to model real data. This might be fixed by building deeper hierarchies of capsule autoencoders ( e. g., complicated scenes in computer graphics are modelled as deep trees of affine-transformed geometric primitives) as well as using input-dependent shape functions instead of fixed templates—both of which are promising directions for future work. It may also be possible to make a much better PCAE for learning the primary capsules by using a differentiable renderer in the generative model that reconstructs pixels from the primary capsules.

Finally, the SCAE could be the 'figure' component of a mixture model that also includes a versatile 'ground' component that can be used to account for everything except the figure. A complex image could then be analyzed using sequential attention to perceive one figure at a time.

## 6   Acknowledgements

We would like to thank Sandy H. Huang for help with editing the manuscript and making Figure 2. Additionally, we would like to thank S. M. Ali Eslami and Danijar Hafner for helpful discussions throughout the project. We also thank Hyunjik Kim, Martin Engelcke, Emilien Dupont and Simon Kornblith for feedback on initial versions of the manuscript.

## Footnotes

[2]This may explain why accessing perceptual knowledge about objects, when they are not visible, requires creating a mental image of the object with a specific viewpoint.

[3]Deriving these matrices from capsule feature vectors allows for deformable objects, see Appendix D for details.

[4]We treat parts as independent and evaluate their probability under the same mixture model. While there are no clear 1:1 connections between parts and predictions, it seems to work well in practice.

[5]We could repeat parts by using multiple instances of the same part capsule.

[6]Discovered objects are *not* used top-down to refine the presences or poses of the parts during inference. However, the derivatives backpropagated via OCAE refine the lower-level encoder network that infers the parts.

[7]This result requires using an additional sparsity loss described in Appendix C; without it the CCAE achieves around $10\%$ error.

[8]We note that we tie the values of the alpha channel $T_m^a$ and the color values $T_m^c$ which leads to better results in the SVHN experiments.

[9]All considered datasets have 10 classes.

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
