[Supplementary Material]

## A    Model Details

We use a convolutional encoder for part capsules and a set transformer encoder (Lee et al., 2019) for object capsules. Decoding from object capsule to part capsules is done with MLPs, while the input image is reconstructed with affine-transformed learned templates. Details of the architectures we used are available in Table 3.

Table 3: Architecture details. S in the last column means that the entry is the same as for SVHN.

| Dataset | MNIST | SVHN | CIFAR10 |
|---|---|---|---|
| num templates | 24 | 24 | 32 |
| template size | $11 \times 11$ | $14 \times 14$ | S |
| num capsules | 24 | 32 | 64 |
| part CNN | 2x(128:2)-2x(128:1) | 2x(128:1)-2x(128:2) | S |
| set transformer | 3x(1-16)-256 | 3x(2-64)-128 | S |

We use ReLu nonlinearities except for presence probabilities, for which we use sigmoids. (128:2) for a CNN means 128 channels with a stride of two. All kernels are $3 \times 3$. For set transformer (1-16)-256 means one attention head, 16 hidden units and 256 output units; it uses layer normalization (Ba et al., 2016) as in the original paper (Lee et al., 2019) but no dropout. We use a 4 layer CNN as the primary encoder with ReLU nonlinearities. All layers have a kernel size of $3 \times 3$ and the last two of them have a stride of 2. Templates are $11 \times 11$ for MNIST and $14 \times 14$ for SVHN and CIFAR10.

For SVHN and CIFAR10, we use normalized sobel filtered images as the target of the reconstruction to emphasize the shape importance. Fig. 6 shows examples of svhn reconstruction and templates. The filtering procedure is as follows: 1) apply sobel filtering, 2) subtract the median color, 3) take the absolute value from the image, 4) normalize for image values to be $\in [0, 1]$

All models are trained with the RMSProp optimizer (Tieleman and G. Hinton, 2012). We run hyper-parameter search on learning rates in the range of .00005 to .0005 and exponential learning rate decay of 0.96 every 10000 or 30000 weight updates. The linear transformation accuracy on a validation set is used as a proxy to select the best hyperparameters.

## B    Attention-based Pooling Encoder

The part object encoder described in Section 2.2 consists of a CNNs followed by attention-based pooling. The intuition that has lead to this design is that it should be possible to instantiate a given part capsule in any place in the image. Therefore, we have a CNN which predicts feature maps of capsule parameters as well as single-channel attention masks for every part capsule. The attention mask is multiplied with the parameter feature map of the corresponding part capsule, which effectively allows to choose parameters from a specific location in the image.

Table 4 contains results of an ablation study, where we use a CNN which is followed by a different kind of a predictor: either a fully-connected layer or $1 \times 1$ convolutions with global average pooling. Changing the type of the part encoder does not affect performance on the original task much but it has significant impact on generalization to novel viewpoints.

Table 4: Ablation of the part capsule encoder. Study conditions are the same as in Section 3.3.

| Method | MNIST | $40 \times 40$ MNIST | AFFNIST |
|---|---|---|---|
| full model | **97.0 (.87)** | **98.5 (.1)** | **92.2 (.59)** |
| linear part enc | 94.8 (3.0) | 98.1 (.26) | 76.3 (2.22) |
| conv part enc | 96.3 (.85) | 97.8 (.95) | 80.1 (2.58) |

Figure 7: Constellation Autoencoder. The set transformer encoder $h^{\mathrm{caps}}$ predicts parameters of two object capsules, which predict affine transformations, precisions and presences of object and part capsules. Finally, input points are explained by a mixture of predictions, where the size of the circle corresponds to its precision.

# C Reconstructions

Figure 6: Caption