[Reviews · NeurIPS 2019]

Reviewer 1



- The proposed model architecture is well validated by empirical comparison to various existing learning architectures as well as 3 benchmark datas, in addition to an ablation study

Reviewer 2



Originality: The method builds on the previous work on capsule networks but is highly original in its own right. The architecture and learning algorithm of the Constellation Autoencoder are a clever innovation that avoids iterative routing while still learning part-whole relations. Related work is well-cited. Quality: The submission is technically sound. The empirical claims are well-supported by a thorough ablation study. The authors are honest about the strengths and weaknesses of the work, though more information on the computational complexity of training relative to other methods would be useful and appreciated. They note challenges in SCA under-performs on CIFAR10 and suggest avenues for improvement. That said, this work feels complete as a foundational paper. Clarity: The paper is well-organized and quite clear given the complexity of the architecture. Those aspects that leave room for interpretation would be remedied were the code made available. Figures 1 and 5 are very well done. Figure 2 would be much easier to parse with white and black / colored foreground. Figures 3 and 4 are too small. Make them full width, moving a figure or other content to the appendix as necessary. Please clarify the difference between the left and right in 4c. L124: “We stop [the] gradient” L136: Missing period. Equation (12) has a typo in the middle. L152: In addition to saying “We find it beneficial to”, it’d be helpful to give the intuition. L244: Clarify what is meant by a “similarity” transformation. L404: svhn in wrong font. Figure 6 has a missing caption. Clarification is sorely needed. Figure 7 should be full width. Significance: The paper is a significant advance for unsupervised learning of object semantics, made especially exciting by SOTA performance on MNIST and SNHN without using labels, data augmentation, ImageNet pre-training, or mutual information. While the method is still far from practical real-world scale and applications, researchers are likely to learn from and build on the method and its component ideas in pursuit of learning with efficiency of generalization closer to that of the brain.

Reviewer 3



The authors propose the Stacked Capsule Autoencoders (SCAE) a novel autoencoding framework based on capsules. It consists of two connected stages: first an Image Capsule Autoencoder (ICAE) extracts part-capsules (including their poses) from an image. Next, a Constellation Autoencoder (CAE) infers object-capsules from these part-capsules by making predictions about their geometrical relationships. ICAE receives an image as input, infers part-capsules (consisting of their poses, presence probabilities, and “special features” that serve as input to the CAE) and is trained to model each pixel in the input image as a mixture of transformed (using the part pose) learned template images (one for each part-capsule). Mixture coefficients are a combination of presence probabilities and an unknown function f_c that reasons about the value of the template at a particular pixel location. CAE receives the special features of part-capsules as input, applies a set transformer to infer K object-capsules (consisting of a feature vector c_k, its presence probability a_k, and a 3 x 3 capsule-camera-relationship matrix (CCR) that represents the affine transformation between capsule and camera. Each object capsule uses a separate MLP to predict a number of part candidates (these do not necessarily correspond to part-capsules) from c_k that each consists of a conditional presence probability a_k, and a 3 x 3 capsule-part-relationship matrix (CPR) that represents the affine transformation between the object capsule and the candidate part. CAE is trained to model each input part-capsule as a mixture over object-capsules and part-candidates, where the part-capsule likelihood explicitly considers the sequence of learned affine transformation (camera to object capsule, object capsule to part) that encourages the system to learn about transformations of otherwise coherent objects. In order to make the full system (SCAE) work, many additional tricks are necessary. Importantly without the use of additional sparsity regularization terms the individual capsules do not specialize as is intended (lines 134-136). These regularization terms rely on 3 limiting assumptions: (1) each image contains only a single object, (2) the number of object classes is known in advance (3) object classes are uniformly distributed across training examples. The first experiment (section 3.1) demonstrate that the CAE is able to group 2d points into constellations, although an additional loss function specific to this very experiment is necessary (lines 155-160). While these results validate the CAE can work as intended, basic sensitivity to the many hyperparameters (especially #object capsules, and #part capsules) is not explored. The second experiment (section 3.2) focuses on the unsupervised classification task on MNIST / CIFAR10 / SVHN. SCA matches state of the art on MNIST and on SVHN using a standard approach (LIN-MATCH) to obtain labels from representations. SCA is significantly worse on CIFAR10, yielding only 25% accuracy using this type of labelling, while representations learned by simply autoencoders yield 31.4% accuracy. These results can be improved by using LIN-PRED to obtain labels, although are still much worse than state-of-the-art 57.6%. The authors argue that the comparison to mutual information based approaches is disadvantageous to them as MI-based approaches rely on data augmentation to estimate MI. I disagree, since SCA incorporates several strong inductive biases about invariances between objects and parts that are not considered in MI-based approaches. Further SCA relies on prior knowledge about the number of classes to formulate sparsity constraints. Finally, an ablation study is performed to assess the importance of several design choices. It is found that the prior sparsity regularizer, noise in object capsules, affine (or similarity transformations), set transformers, and special features are all necessary to obtain the reported results on MNIST. It is further confirmed that SCA is still able to generalize to out of distribution viewpoints that is due to the incorporated inductive biases. ####### Overall I find the proposed capsule framework quite original, although I find the quality and clarity of the paper somewhat lacking (see detailed comments below). Moreover, based on the current results and the amount of engineering involved I do not consider this contribution particularly significant yet in its current form. The experimental results clearly demonstrate that SCA relies heavily on specific sparsity regularizers that make 3 very limiting assumptions (outlined above). Further, while good results are obtained on MNIST / SVHN, the poor results on CIFAR10 are a clear indicator that work remains to be done. Given the amount of engineering that is currently incorporated already I do not believe that solving these challenges will be a matter of incorporating yet another regularizer. Rather, it is likely that the system is not fully understood yet. Further, I would encourage the authors to explore other metrics than the ones proposed in this paper. The view of capsules introduces an invariance that should be reflected in terms of out-of-distribution generalization (not just to novel viewpoints, but also novel objects by combining new parts). Secondly, an invariance also implies better learning efficiency, as experiences from different samples (different viewpoints of the same object) can be related to each other. Further, one of the original strengths of capsules was multi-object representation, which are not explored here but could be interesting also. Finally, there are several important hyper-parameters that were left unexplored in this paper, which are essential in determining the generality of the proposed framework: in particular the reliance on an approximately correct number of part-capsules / object-capsules, and candidate parts. Others are mentioned below. ######## Detailed comments: General: I would encourage the authors to remove the symbols for part / camera / whole. The introduction (line 31) cites two rather old works (Rock, 1973; Hinton 1979) as evidence that human vision relies on coordinate frames. It would be good to incorporate also more recent work that is representative of our current understanding of human vision and coordinate frames. In the introduction (lines 37-41) it is not clear to me that the enumeration (i) - (iii) all follow from the fact that each part makes a separate prediction for a whole, which is assumed to be present when predictions of many parts agree. For example (i), why should parts not take place in multiple wholes (i.e. to facilitate hierarchical/modular concepts)? (ii) seems like a technicality, why is this a strict restriction? (iii) Likewise, why is learning strictly supervised in this case? Arguably the work from Rawlinson et al. (2018) already shows that this is not necessary. In Section 2.1 (line 82-83) why can an object capsule only represent one object at a time? As far as I can tell there is nothing that prevents an object capsule from representing the points belonging to multiple objects at this stage, i.e. there is no notion of object yet. In Section 2.1 (line 87) footnote 2 suggests that deriving the CPR matrices from c_k would allow for deformable objects. Here CPRs are modeled as “the sum of an input-dependent component and a constant bias”. What exactly is meant with this? Since the input to the part-MLP is c_k, does that mean that deformable objects are possible in the current setting? Further it is mentioned that object-capsules are encouraged to specialize by using an L2 penalty. How exactly is an L2 penalty used to achieve this? In Section 2.1 (line 93) it is unclear how the n predicted parts factor in assigning part-capsules to object capsules. For every object capsule k there are n predicted parts that each yield a different part likelihood p(x_m | k, n), yet the computation of k-star only considers a single n. Is the argmax taken w.r.t. all predicted parts for a given object capsule? This is not clear from the text (or footnote 3) although perhaps this is meant with “we treat parts as independent and evaluate their probability under the same mixture model”. I also wonder what the implications are of this model if N < M as seems possible (line 84)? In Section 2.2 (line 115) pixels are modelled as independent. This is completely unrealistic for any real-world image, and it should be commented on what the implications are of this assumption here. Why not consider a spatial mixture model, as has been shown to work well in modeling objects in images (Greff et al. 2017)? In Section 2.2 (line 118) I find the role of the function f_c confusing. Footnote 4 suggests that it is meant to have templates participate that have a non-zero value at a pixel location, however the converse is also true: f_c essentially does not allow a template to participate in reconstructing a pixel, unless it has a non-zero values (I am assuming that f_c is a linear function of the pixel-values, as I can not find an exact definition somewhere). While this may not be so problematic under the naive assumption that pixels are independent, this is not the case for a more realistic setting in which a particular template that spans multiple pixels may yield a zero-value for a particular pixel. Section 2.2 (line 120) footnote 5. Why are discovered objects not used top-down to refine the presence or poses of the parts during inference? Currently inference of part-capsules is completely bottom-up, whereas arguably inferring (parts belonging to) objects requires both bottom-up and top-down inference. I understand that this is an assumption made here, yet it is fundamentally a limitation that should be discussed. In Section 2.3 (lines 144-150) several assumptions are made. It is assumed that each image contains only a single object, that the number of classes is distributed uniformly across the dataset, and finally that the number of classes is in fact known. While none of these assumptions are expected to hold in a more realistic setting, I have mostly an issue with assuming the number of classes. This is a form of supervision, and essentially imposes a pre-specified notion of “object” on the system. If these regularizers are indeed so important to get the system to work as is suggested (lines 134-136), then in the very least they should not depend on external information of this kind. Perhaps a better regularizer would be to encourage each image to be represented by only a single object capsule. In Section 2.3 (lines 155 - 160) it is not clear why this loss is required for the stand-alone constellation model experiments on point data? Is the strong L2 regularizer that was referred to previously to encourage object-capsules to specialize? Section 3.1: it would be instructive to evaluate the Constellation Autoencoder in some more detail. How does correctly choosing the number of object capsules in relation to the number of ground-truth constellations in the data affect performance? How does correctly choosing the number of parts? I don’t find the comparison to a simpler decoder particularly useful, rather a comparison to similar unsupervised instance segmentation algorithms based on mixture models (Greff et al. 2017; 2019) would be much more informative. Although these typically model pixels, the 2d points (or more generally - parts) exhibit similar spatial structure that should allow a comparison to be made. In Section 3.1 (lines 204-216), it is not clear why all these different methods of obtaining a classification for a given image are considered. From the text it appears that LIN-MATCH is the default used in the literature, and I assume that for the sake of comparison this is also the method considered for all baselines in Table 1? Section 3.1: (line 227) I would say that SCA underperforms significantly on CIFAR10. If I focus on the LIN-MATCH row, then the results that are obtained are even worse than training a simple autoencoder (AE) on images for reconstruction, followed by the exact same clustering procedure. While the best-performing approaches apply data-augmentation this is not so different from the strong inductive biases that are inherent to capsules. Secondly, SCA makes use of task-specific regularization that considers the true number of classes during training time. Especially based on the AE result I think that it is reasonable to conclude that there is almost no benefit in using capsules on this task. Section 4: (lines 262-268) while I understand that routing is expensive, the lack of routing in this paper is not strictly an advantage. For example, currently there is no top-down interaction during inference, which is arguably a desirable property in many settings. It would be good to also mention the flipside of this in discussing related work. Section 4: (lines 274-276) it is not necessarily true that unsupervised instance segmentation approaches based on spatial mixture models eg. Greff et al. (2017, 2019) are unable to group together parts to form objects. If one were to apply IODINE on single digit MNIST images it seems natural that it would discover parts as the “objects” with which to represent the scene. Section 4: I would encourage the authors to keep the related work section on the one hand more focused, and on the other hand more detailed by incorporating not only the most recent prior work. For example, Mutual Information based approaches to representation learning are not particularly related to the proposed framework, i.e. compared to say geometric approaches. Secondly, the work by Burgess et al. (2019) is cited yet this is only a technical report. Meanwhile earlier work on unsupervised instance segmentation (as presented in Greff et al. 2019 or Burgess et al. 2019) is ignored and should be added. ##################### POST REBUTTAL ########################## I have read the author response and the other reviews, and remain of the opinion that this work is currently not suitable for publication. I appreciate that the authors responded to many of my detailed comments in their response, and also that they acknowledge that the proposed system does not work on CIFAR10. In my view, the original submission did not do a good job at acknowledging the limitations of the proposed system, and going forward I would encourage the authors to incorporate a discussion of these limitations. That said, based on the results presented on CIFAR10 I can only conclude that the system in its current form (although novel and interesting) is not well understood. This, plus the fact that the system is highly engineered and incorporates many design choices specific to the datasets considered leads me to believe that the work proposed here is not yet ready for others to build upon, and therefore not ready for publication. The authors promised several significant updates in their response, including a new background model on CIFAR10, but unfortunately these can not be evaluated at this time. Solely relying on the goodwill of the authors to significantly improve the paper at publication time would not set the right precedent for publication at NeurIPS.

[Author Response · NeurIPS 2019]

We thank the reviewers for their constructive feedback and we address their concerns directly.

**Review 1.** We are going to incorporate your suggestions in the next version of the paper. Moreover: 1. Special features are used as an input to the CAE; while they do not have any imposed meaning, they provide information about the image to CAE and allow adapting the ICAE encoder to the needs of CAE. 2. Sparsity is imposed only on the CAE level. 3. Details on the solver are provided in the appendix. 4. Since different aspects of the model considered in the ablation study are not additive, providing a cumulative performance decrease of combined ablations would require rerunning experiments. We will consider adding these results to the next version of the paper.

**Review 2.** Our method is feed-forward without iterations, so it's O(1) in that sense; it's O($n^2$) in the number of capsules n, since the number of model parameters grows quadratically with n, similarly to the previous versions of capsule networks. We are going to release the code in due time.

**Review 3.** General comments: 1. We focus on modeling single objects in order to simplify and better understand the method. In future work, we will extend SCA with deeper hierarchies of transformations, where higher levels handle separate objects. 2. Regarding other metrics, we are going to run few-shot classification and meta-learning studies as future work. This will also validate whether our method learns in a more data-efficient manner compared to other methods. 3. While not included in the paper, our results show that SCA is not sensitive to the number of object or part capsules, and this holds for both constellation and image experiments. We will add this analysis. 4. We think that strong inductive biases are desirable (e.g. CNN). Mammals rely on them and they are a major motivation of this work. Data augmentation helps MI-based methods but it is statistically inefficient. 5. We removed the symbols for camera/part/object; we are unaware of references for coordinate frame usage in human vision that are more relevant. We removed the list of drawbacks of previous capsule architectures from the introduction.

**Section 2.1:**
6. Deformations are currently allowed and modelled as $CPR_{k,n} = CPR_{k,n}^b + f_n^{CPR}(c_k)$; the first term is a bias and the second term is input dependent. We add $\alpha||f_n^{CPR}(c_k)||_2^2$ to the final loss, which discourages input-dependent transforms. 7. Since deformations are penalized, the model tries to not use them. Explaining multiple objects with a single capsule would require severe deformations if the objects appear in different configurations. 8. line 93: the $\arg\max$ in Eq. 5 should be over both $k$ and $n$. Every object capsule predicts $N \neq M$ parts and we need to choose $M$ parts out of all $NK$ predictions. We only require that $M <= NK$.

**Section 2.2**:
9. We do use a spatial mixture model as in Greff et al. 2017, which does model pixels as *conditionally* independent. Many VAEs and all approaches that consider mean-squared error in the pixel space make the same assumption. 10. $f_c$ is linear; it was developed for MNIST and does not generalize to other datasets, which might also cause limited performance on CIFAR10. We will explain it in the paper and investigate better ways of specifying mixing probabilities. 11. We currently do not use discovered objects to refine parts. Instead, we rely on amortized inference, where the encoder can learn to mimic the EM procedure from previous capsules. In principle, we could run CAE iteratively, where at every iteration we encode parts and then reconstruct them. Our initial experiments resulted in unstable training, and we will investigate this in future work.

**Section 2.3**:
12. We do use the true number of classes as a hyperparameter for the sparsity regularizer, but this value need not be known; it can be fitted on the validation set instead. It is a good idea to encourage just one active object capsule per input image and we will try it. The notion of uniformly distributed classes across training data motivated one regularizer, but it is not strictly necessary. We will reformulate this in the paper. 13. The additional cost function in the constellation experiment will be removed as the same regularizers as used in image experiments are sufficient to obtain the reported performance.

**Section 3.1**: 14. we note that our model with simpler decoder is similar to Greff et al. 2017; 2019, but where inference is amortized instead of performing EM as in Greff et al. 2017 or gradient ascent in the latent space as in Greff et al. 2019. What makes our model different is the explicit geometrical structure of the decoder.

**Section 3.2**:
15. LIN-MATCH is used in the ablation study. In retrospect, we feel that only LIN-MATCH and LIN-PRED provide useful information. The other two will be removed from the paper. 16. It is fair to conclude that SCA failed on the Cifar10 experiment. We just developed a mixture of a capsule model with a background model tto deal with clutter and will update the paper.

**Section 4**:
17. We will add the drawbacks of having no routing to this section. We note that iterative refinement of part-capsule assignments is possible with our approach, but has not been explored yet. 18. While it is true that MONet or IODINE would discover parts if applied to single-object data, there is no clear way to impose hierarchy. In our work, it is possible to stack multiple levels of object capsules, in which case we can have more levels of decomposition. This is an area of future work. We will reformulate this paragraph. 19. We are going to make this section more focused. We note that citing technical reports published on arxiv is standard practice.

[Meta-Review · NeurIPS 2019]

After reading the authors feedback, the reviewers were mostly in agreement about the merits of this work (originality, potential significance) and limitations (highly engineered, not perfectly understood). Overall, this is considered as being a paper worth presenting at NeurIPS. However, It seems that authors do not appear fully committed to releasing code, which is something that the NeurIPS PC strongly encourage. This will be very useful also to get this paper read and cited by peers.